# Intrathecal morphine exacerbates paresis with increasing muscle tone of hindlimbs in rats with mild thoracic spinal cord injury but without damage of lumbar α-motoneurons

**Katsuhiro Kawakami, Satoshi Tanaka**[ID]*, **Yuki Sugiyama, Noriaki Mochizuki, Mikito Kawamata**[ID]

Department of Anesthesiology and Resuscitology, Shinshu University School of Medicine, Nagano, Japan

* s_tanaka@shinshu-u.ac.jp

**Data Availability Statement:** All relevant data are within the paper and its Supporting Information files.

## Abstract

Adverse effects of morphine on locomotor function after moderate to severe spinal cord injury (SCI) have been reported; however, the effects after mild SCI without damage of lumbar α-motoneurons have not been investigated. We investigated the effects of lumbar intrathecal morphine on locomotor function after mild thoracic SCI and the involvement of classic opioid receptor activation. A mild thoracic contusive SCI was induced in adult rats at the T9-T10 spine level under sevoflurane anesthesia. We evaluated the effects of single doses of intrathecal morphine and selective μ-, δ-, and κ-opioid receptor agonists, continuous infusion of intrathecal morphine for 72 hours, and administration of physiological saline on locomotor function and muscle tone in the hindlimbs. The numbers of damaged and total α-motoneurons in the lumbar spinal cord were also investigated. Single doses of morphine aggravated residual locomotor function after SCI but did not affect functional recovery. Single doses of morphine and μ- and δ-opioid receptor agonists significantly aggravated residual locomotor function with increases in muscle tone after SCI, and the effects of the drugs were reversed by naloxone. In contrast, continuous infusion of morphine led to persistent decline in locomotor function with increased muscle tone, which was not reversed by naloxone, but did not increase the number of damaged lumbar α-motoneurons. These results indicate that a single dose of morphine at an analgesic dose transiently increases muscle tone of the hindlimbs via activation of spinal μ- and δ- opioid receptors, resulting in further deterioration of locomotor function in the acute phase of mild SCI. Our results also suggest that an increased dose of morphine with prolonged administration leads to persistent decline in locomotor function with increased muscle tone via mechanisms other than direct activation of classical opioid receptors. Morphine should be used cautiously even after mild SCI.

## Introduction

Spinal cord injury (SCI) frequently leads to alterations in ambulatory and many other bodily functions that affect the quality of life. Opioids, such as morphine and fentanyl, have been

**Funding:** This work was supported by grants-in-aid for scientific research from the Ministry of Education, Culture, Sports, Science and Technology, Tokyo, Japan (Grant No.21592002 to KK). https://www.jsps.go.jp/j-grantsinaid/ The funders had no role in study design, data collection and analysis, decision to publish, or preparation of the manuscript.

**Competing interests:** The authors have declared that no competing interests exist.

commonly used as analgesics for the treatment of acute pain arising from not only damage to musculoskeletal structures but also from surgical injuries for spine decompression and stabilization in patients with SCI [1]. However, experiments using an SCI rat model and clinical observations in humans indicated that intrathecal (IT) morphine at an analgesic dose exacerbates motor dysfunction initiated by spinal cord ischemia [2–4]. Furthermore, in rats with moderate to severe contusive SCI, IT morphine increases the lesion size at the epicenter and negatively affects functional recovery [5–7]. Thus, morphine at analgesic concentrations may affect motor function after various types of SCI.

In previous studies in which ischemic SCIs were produced by occlusion of the descending aorta [2–4, 8] or traumatic SCIs were produced by contusion at the T12-13 spine level [5–7], it was shown that morphine-induced aggravation of locomotor function may be partially due to some damage of the lumbar spinal cord near the epicenter of the SCI. Effects of morphine on animals with almost no residual locomotor function after moderate to severe SCI were also investigated in previous studies. However, it has not been clarified whether morphine affects residual locomotor function after a mild degree of SCI and whether morphine aggravates the residual motor function of the lower extremities in individuals with cervical or thoracic SCI in a condition in which α-motoneurons in the lumbar spinal cord are considered to be intact in the acute phase. Clinically, there is wide individual variation in the severity of symptoms and functional recovery varies depending on initial severity and subsequent treatment [9]. It is clinically essential to investigate the safety of morphine use for individuals with various types of SCI in order to promote early rehabilitation.

This study was therefore designed to investigate whether IT morphine leads to aggravation of locomotor function, what kind of changes in motor tone occur, and whether activation of spinal opioid receptors is involved in them in the early phase of mild contusive thoracic SCI distant from the lumbar region, which is a clinically common traumatic SCI [10]. Morphine is given both as a single dose and as continuous infusion in a clinical setting. Accordingly, the effects of both single and continuous IT morphine administration were examined.

## Materials and methods

### Animals

All of the protocols of this study were approved by the Animal Care and Use Committee of Shinshu University School of Medicine (number 190151). Animals were treated in accordance with the guidelines of the National Institutes of Health. Adult male Sprague-Dawley rats, 8–9 weeks old (weighing 240–300 g), were obtained from Nippon SLC (Tokyo, Japan) and housed in $40 \times 60 \times 30$ cm plastic cages with soft bedding under a 12:12 h day: night cycle at 22-24˚C. Each animal was separated and given water and food pellets ad libitum. Every effort was made to reduce the number of animals used in this study.

### Intrathecal catheter implantation for drug delivery

The rats were anesthetized with sevoflurane (5.0% initially and maintained with 3.0–4.0%) in oxygen delivered via a face mask. A PE-10 polyethylene catheter (Becton Dickinson Primary Care Diagnostics, Sparks, MD, USA) was inserted 15 mm cephalad into the lumbar subarachnoid space at the L4-L5 intervertebral space. The catheter was then tunneled subcutaneously to emerge at the neck. The wound at the lumbar region was sutured in layers. Rats showing motor weakness or paralysis of lower limbs were excluded from the study. Lidocaine (10 μg) (Sigma, St. Louis, MO, USA) was injected to confirm the location of the catheter. Rats that were not paralyzed by lidocaine were excluded from the study (n = 3). Penicillin G (100,000 units/kg) (Sigma, St. Louis, MO, USA) was administered subcutaneously every 2 days after

surgery throughout the experiments. The drugs remaining in the catheter were always washed out by 15 µl of saline.

## Spinal contusion injury

At least 6 days after implantation of the catheter, a T9-T10 laminectomy was performed under sevoflurane anesthesia. The thoracic vertebrae were immobilized with a stereotaxic instrument. SCI was produced using the New York University impactor device (New York University, New York, NY, USA) [11] by dropping a 10-g impactor from a height of 10 mm onto the exposed spinal dura mater. After the injury, the wound was sutured in layers and the animals were returned to clean cages. The animals in the sham-operated group underwent IT catheter insertion and laminectomy at T9-T10 but not SCI. After the surgeries, general activity of the rats was monitored every day. Each rat's bladder was manually expressed twice daily until voiding was re-established and then once a day for the duration of the study. Ringer's solution (5 ml) was injected subcutaneously once a day. All surviving rats on day 14, as well as all animals reaching ethical endpoints during the experiment, were euthanized with sevoflurane excess inhalation. Ethical endpoints were defined as lethargy and self-mutilation of any body parts. During the experiment, five rats that received continuous infusion of morphine after thoracic SCI met the humane endpoints. Three of the rats were lethargic and two displayed self-mutilation. It has been reported that morphine administration leads to an increase the incidence of autophagia and weight loss in animals with SCI compared to those without morphine [6]. Data from those animals were excluded from data analysis.

## Assessment of locomotor function and muscle tone

Locomotor function was assessed using the Basso, Beattie, and Bresnahan (BBB) locomotor rating scale in open fields (120 × 120 cm) [12]. The rats were acclimated to observation fields for 5 minutes each day for 3 days prior to the SCI. This scale ranges from 0 (= no observable hindlimb movement) to 21 (= normal locomotion).

Muscle tone of the hindlimbs was assessed using the Ashworth scale to distinguish between flaccid paralysis and increases in muscle tone [13]. The head of each rat was held gently and passive movements of the hindlimbs were made by the observer after the rat was completely relaxed. Resistance against the passive movement (knee flexor and extensor muscles) was evaluated by the Ashworth scale as follows: scale 0 indicates "no increase in tone", scale 1 indicates "slight increase in tone giving a 'catch' when the limb is moved in flexion or extension", scale 2 indicates "more marked increase in tone, but the limb is easily flexed ", scale 3 indicates "considerable increase in tone-passive movement difficult", and scale 4 indicates "limb rigid in flexion or extension" [14]. The BBB and Ashworth scores were determined by blinded investigators.

## Histological analysis of α-motoneurons in the lumbar spinal cord

The rats were perfused intracardially with heparinized saline followed by 4% cold and buffered paraformaldehyde after sevoflurane anesthesia. Transversal sections of 5 µm in thickness were obtained by cutting the spinal cord at the level of the lumbar enlargement and were stained by the Klüver-Barrera method to assess neuronal damage [15]. For systematic analysis, 5–8 sections of each spinal cord, collected every 400 µm, were examined. An α-motoneuron with a visible nucleus was defined on the basis of a diameter larger than 25 µm and location in lamina IX [16]. Dark-stained α-motoneurons were assessed as damaged neurons by a blinded observer. The number of α-motoneurons in the left ventral horn of the lumbar spinal cord was counted and the ratio of damaged α-motoneurons to total α- motoneurons was calculated in each animal.

## Experimental design

**Experiment 1: Effects of a single dose of IT morphine on locomotor function after SCI.** A single dose of 30 μg of morphine (morphine hydrochloride; Daiichi Sankyo, Tokyo, Japan) in 10 μl of physiological saline (morphine group), or 10 μl of physiological saline (saline group) was administered through the IT catheter 6 hours after SCI. Thirty μg of morphine was determined according to a previously described dose that has a sufficient antinociceptive effect to nociceptive stimuli by heat and electrical shock in rats with moderate thoracic SCI [6].

**Experiment 2: Effects of single doses of IT morphine and selective μ-, δ-, and κ-opioid receptor agonists on locomotor function and muscle tone 6 hours after SCI.** μ ([$_D$-Ala$^2$, N-Me-Phe$^4$, Gly$^5$-ol]-enkephalin acetate salt: DAMGO), δ ([$_D$-Pen$^{2,5}$]-enkephalin hydrate: DPDPE), and κ (±-trans-U-50,488 methanesulfonate salt: U50,488H) selective opioid receptor agonists were obtained from Sigma (St. Louis, MO, USA). Morphine (3, 10, or 30 μg/10 μl), DAMGO (1, 3, or 10 ng/10 μl), DPDPE (10, 30, or 100 μg/10 μl), or U50,488H (15, 45, or 150 μg/10 μl) was administered through the IT catheter 6 hours after the SCI or after the sham operation without SCI, and the BBB and Ashworth scores were evaluated at 30 min after administration of each drug. Naloxone (60 μg/15 μl) (Sigma, St. Louis, MO, USA), an opioid receptor antagonist, was then intrathecally injected to the rats with decrease in the BBB scores at the highest doses of the drugs (30 μg morphine, 10 ng DAMGO, 100 μg DPDPE, and 150 μg U50,488H), and the BBB and Ashworth scores were recorded again at 30 min after administration of naloxone.

**Experiment 3: Effects of continuous infusion of IT morphine for 72 hours on locomotor function and muscle tone.** All catheterized rats were randomly divided into four groups as follows: 1) contused rats that received continuous IT morphine, 2) contused rats that received continuous IT physiological saline, 3) sham-operated rats without SCI that received continuous IT morphine, and 4) sham-operated rats without SCI that received continuous IT physiological saline. In these 4 groups, six hours after the SCI or sham operation, a single dose of morphine (30 μg/10 μl) or physiological saline (10 μl) was administered through the IT catheter followed by continuous infusion of morphine (3 μg/μl/hr) or physiological saline (1 μl/hr), and the BBB and Ashworth scores were evaluated 30 min after the start of continuous infusion of the drugs. This continuous infusion dose of IT morphine was chosen according to a previous study in which that dose had a sufficient antinociceptive effect in tail-flick and colorectal distension tests [17]. Continuous infusion was performed for 72 hours using a subcutaneously implanted osmotic pump (Alzet micro-osmotic pump, model 1003D. DURECT Corporation, Cupertino, CA, USA). Under sevoflurane anesthesia, the pump was removed at 14 days after the start of continuous infusion. Then it was confirmed that there was no residual solution in the pump.

Some rats in each group 3 days after SCI or sham surgery were used for histological analysis of α-motoneurons in the lumbar spinal cord. Responses to IT naloxone (60 μg) were also examined at 30 min after the end of 72-hour continuous IT administration of morphine to rats with SCI. A single dose of 30 μg of IT morphine was administered to rats with SCI at 30 min after the end of 72-hour continuous infusion of physiological saline. After another 30 min, sixty μg of IT naloxone was administered. The BBB and Ashworth scores were evaluated at 30 min after administration of each drug.

## Statistical analysis

The sample size was determined on the basis of a previous study in which locomotor function of rats with SCI was evaluated by using the BBB rating scale [18]. Data for the BBB score are expressed as means ± standard deviation (SD). Data for the Ashworth score are presented as

medians with first and third quartiles. Mann-Whitney's U test or the Kruskal-Wallis rank test followed by Dunn's multiple comparison test were used to compare data for the BBB and Ashworth scores between the different groups. The ratio of dark-stained to total number of α-motoneurons and the total count of α-motoneurons are expressed as means ± SD. One-way ANOVA followed by Scheffe's F-test were used to analyze differences in the number of α-motoneurons and the ratio of injured α-motoneurons to total α-motoneurons. SPSS software version 24.0 (IBM Japan, Ltd., Tokyo, Japan) was used for statistical analysis. A significant difference was defined as $P < 0.05$.

## Results

### Effects of a single dose of IT morphine on locomotor function after SCI

SCI at the T9-T10 spine level led to locomotor dysfunction but not complete paralysis of the hindlimbs. There were no significant differences in the BBB scores between the groups at 6 h after SCI before morphine administration ($P = 0.085$). The BBB score of the morphine group at 30 min after administration (6.5 hours after SCI) was significantly lower than that of the saline group ($P = 0.006$). The locomotor function aggravated by morphine (IT, 30 μg) returned to a level similar to that in the saline group on day 1. There was no significant difference in subsequent functional recovery between the groups (Fig 1).

### Effects of single doses of morphine and opioid receptor agonists on residual locomotor function and muscle tone after SCI

IT administration of 30 μg of morphine ($P = 0.004$), 10 ng of DAMGO ($P = 0.001$), and 100 μg of DPDPE ($P = 0.04$) resulted in further aggravation of residual locomotor functions after SCI. These reductions of BBB scores were in a dose-dependent manner (Fig 2A–2C). The effects of these drugs were reversed by IT naloxone. In contrast, none of the doses of U50,488H tested in this study affected locomotor function after SCI, and therefore, naloxone was not used in this group (Fig 2D).

Mild SCI did not change the Ashworth scores of the hindlimbs of rats as shown in Fig 2E–2H. IT administration of 10 μg or more of morphine (10 μg, $P = 0.017$), 10 ng or more of DAMGO (10 ng, $P = 0.029$), and 30 μg or more of DPDPE (30 μg, $P = 0.013$) increased the Ashworth scores 6 hours after SCI. The effects were in a dose-dependent manner and were reversed by IT naloxone (Fig 2E–2G). U50,488H did not affect the Ashworth score (Fig 2H). Thirty μg of morphine, 10 ng of DAMGO, 100 μg of DPDPE, and 150 μg of U50,488H did not affect the BBB and Ashworth scores in sham-operated rats without SCI (S1 Fig). The decrease in the BBB score after intrathecal administration of morphine, DAMGO, and DPDPE were clearly associated with the increase in the Ashworth score. The recovery of the BBB score after intrathecal administration of naloxone was related to reduction of the Ashworth score (S2 Fig).

### Effects of continuous infusion of IT morphine on locomotor function and muscle tone after SCI

Continuous infusion of IT morphine for 72 hours significantly aggravated locomotor function and delayed locomotor functional recovery of rats after SCI compared with those in rats that received IT saline ($P = 0.002$) but did not change locomotor function of the sham-operated rats without SCI (Fig 3A). Continuous infusion of IT morphine to rats with SCI significantly increased the Ashworth scores up to 4 days compared with the Ashworth scores in rats that received IT saline ($P = 0.002$) (Fig 3B).

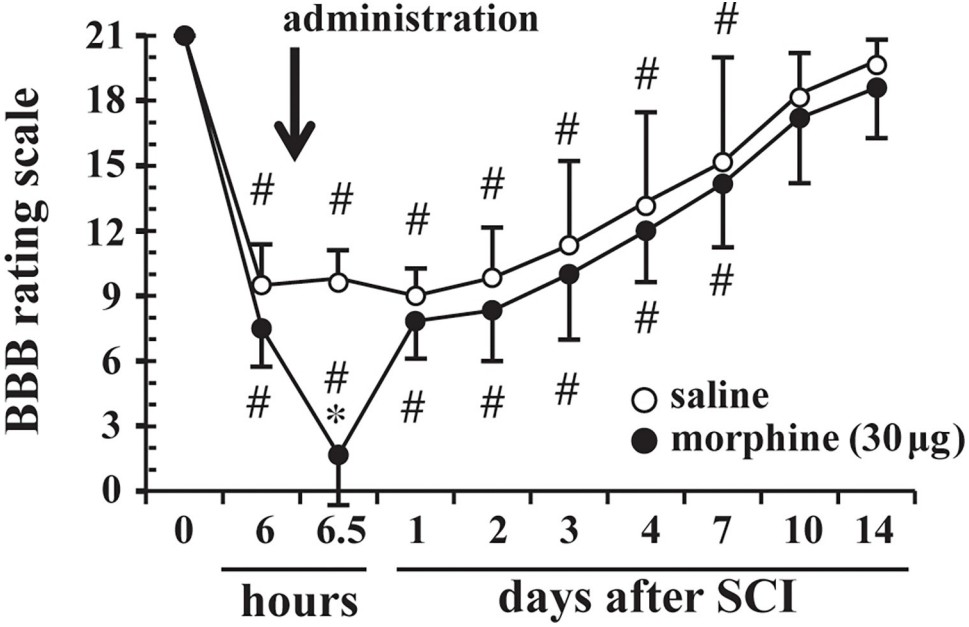

**Fig 1. Effects of a single dose of intrathecal (IT) morphine on residual locomotor function after mild thoracic spinal cord injury (SCI).** Locomotor function was measured by the 21-point Basso, Beattie, and Bresnahan (BBB) locomotor rating scale (0 = complete paralysis; 21 = normal locomotion) before SCI (time 0) and after administration of IT morphine (30 μg/10 μl) or IT normal saline (10 μl). IT morphine or IT saline was administered 6 hours after SCI. An arrow indicates the timing of administration of drugs (morphine or saline. n = 6 in each group. Data are presented as means ± SDs. * $P < 0.05$ compared to saline. # $P < 0.05$ compared to time 0.

### Effects of naloxone on locomotor function 72 hours after SCI

The sustained locomotor dysfunction induced by continuous infusion of IT morphine could not be reversed by IT naloxone 72 hours after SCI (Fig 4A). A single dose of 30 μg of IT morphine after 72-hour continuous infusion of normal saline to rats with SCI aggravated the residual locomotor function, and the effect was reversed by IT naloxone (Fig 4B).

### Viability of α-motoneurons in the lumbar spinal cord after continuous infusion of morphine for 72 hours after SCI

Although the analysis revealed the occasional presence of dark-stained (considered to be damaged) α-motoneurons, there was no significant difference between the groups in the total number of α-motoneurons ($P = 0.104$) or the ratio of dark-stained α-motoneurons to normal α-motoneurons ($P = 0.642$) (Fig 5).

### Discussion

To the best of our knowledge, this is the first study showing the effects of morphine on residual locomotor function of the lower extremities controlled by α-motoneurons of the lumbar spinal cord in rats with mild thoracic SCI. The major findings of this study were as follows: (1) a single dose of morphine (30 μg) at an analgesic dose transiently aggravated locomotor function but did not affect subsequent recovery (Fig 1), (2) IT administration of μ- and δ- opioid receptor agonists increased muscle tone and aggravated locomotor function in the early phase of thoracic SCI, and the effects were reversed by naloxone (Fig 2), (3) continuous infusion of morphine for 72 hours caused a persistent decline in locomotor function with increases in muscle tone, and the effects were not reversed by naloxone (Figs 3 and 4), and (4) continuous

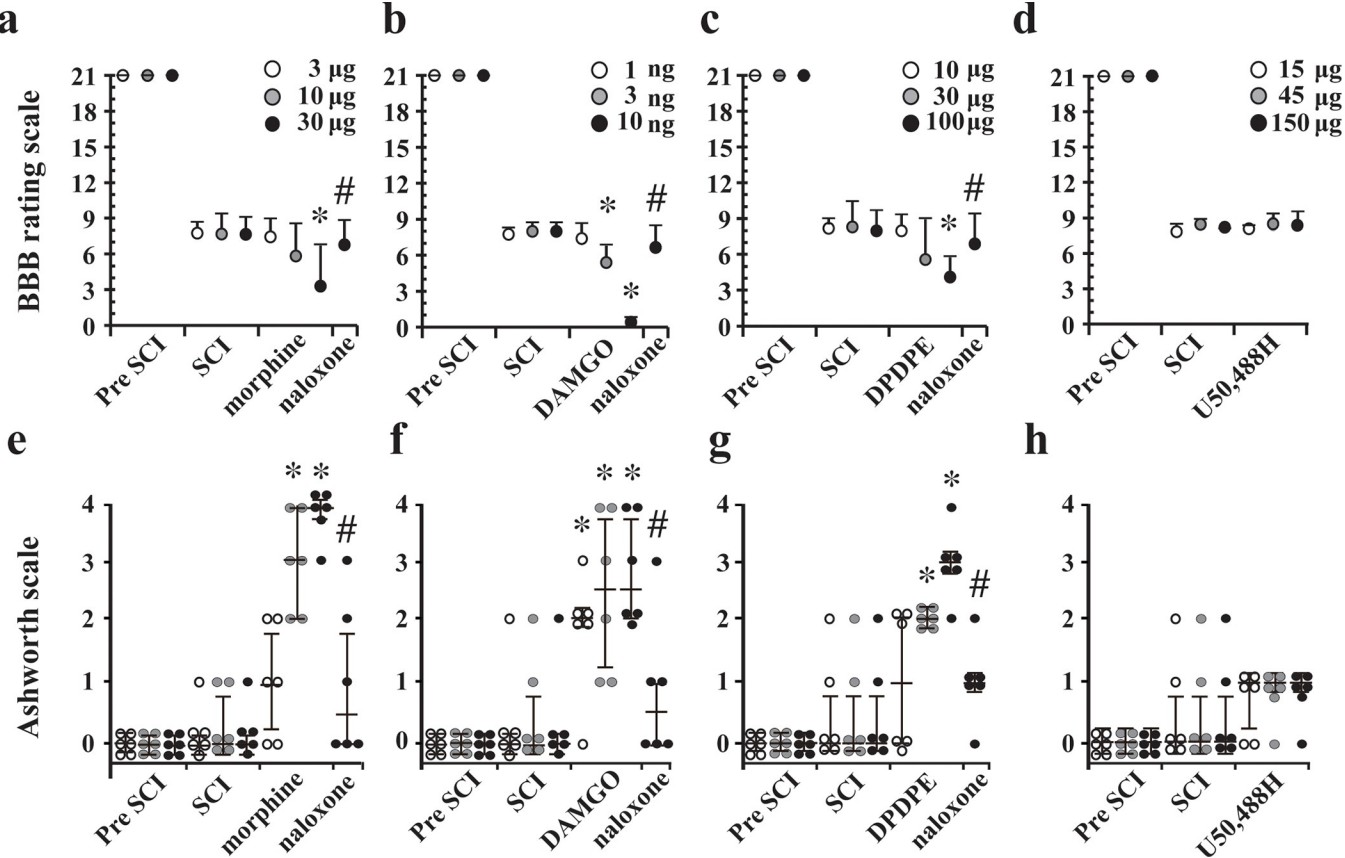

**Fig 2.** Locomotor function (a-d) and muscle tone (e-h) after intrathecal (IT) administration of morphine and µ- (DAMGO), δ- (DPDPE), and κ- (U50,488H) selective opioid receptor agonists after mild thoracic spinal cord injury (SCI). Locomotor function was evaluated using the 21-point Basso, Beattie, and Bresnahan (BBB) locomotor rating scale (0 = complete paralysis, 21 = normal locomotion) before SCI (pre SCI), 6 hours after SCI (SCI), after administration of each drug, and after IT administration of naloxone (60 µg). The panels show effects of (a) morphine, (b) DAMGO, (c) DPDPE, and (d) U50,488H. Data of the BBB score are presented as means ± SDs. The Ashworth scales (0 = no increase in tone; 4 = increase in muscle tone) were used to evaluate changes in muscle tone of the hindlimbs before SCI (pre SCI), 6 hours after SCI (SCI), after administration of each drug, and after administration of IT naloxone (60 µg). The panels show effects of (e) morphine, (f) DAMGO, (g) DPDPE, and (h) U50,488H. Data are presented as scatter dot plots of the Ashworth score displaying the median as a line and the 25–75 percentiles. n = 6 in each dose. * $P < 0.05$ compared to post SCI. # $P < 0.05$ compared to before naloxone.

infusion of morphine did not increase damaged α-motoneurons in the lumbar spinal cord compared with those in the saline group (Fig 5). Taken together, these results suggest that morphine-induced aggravation of locomotor function after thoracic SCI is, at least in part, attributed to hypertonia of the hindlimb muscles via activation of µ- and/or δ- opioid receptors but not κ-opioid receptors. Our results also suggest that the mechanism of morphine-induced aggravation of locomotor function varies depending on the duration of morphine administration in the early phase of SCI. It has been reported that a single dose of 90 µg morphine delays locomotor functional recovery after moderate SCI [6]. Therefore, locomotor functional recovery from SCI may vary depending on the dose of morphine and duration of morphine administration.

Morphine, DAMGO and DPDPE, but not U50,488 H, increased the Ashworth scores and the effects were reversed by IT naloxone (Fig 2E–2H), suggesting that activation of µ- or δ-opioid receptors causes an increase in muscle tone, resulting in further aggravation of locomotor function. The results are similar to the results of a previous study showing that activation of µ- or δ-opioid receptors induces spastic paralysis in an ischemic model of SCI [4]. The IT

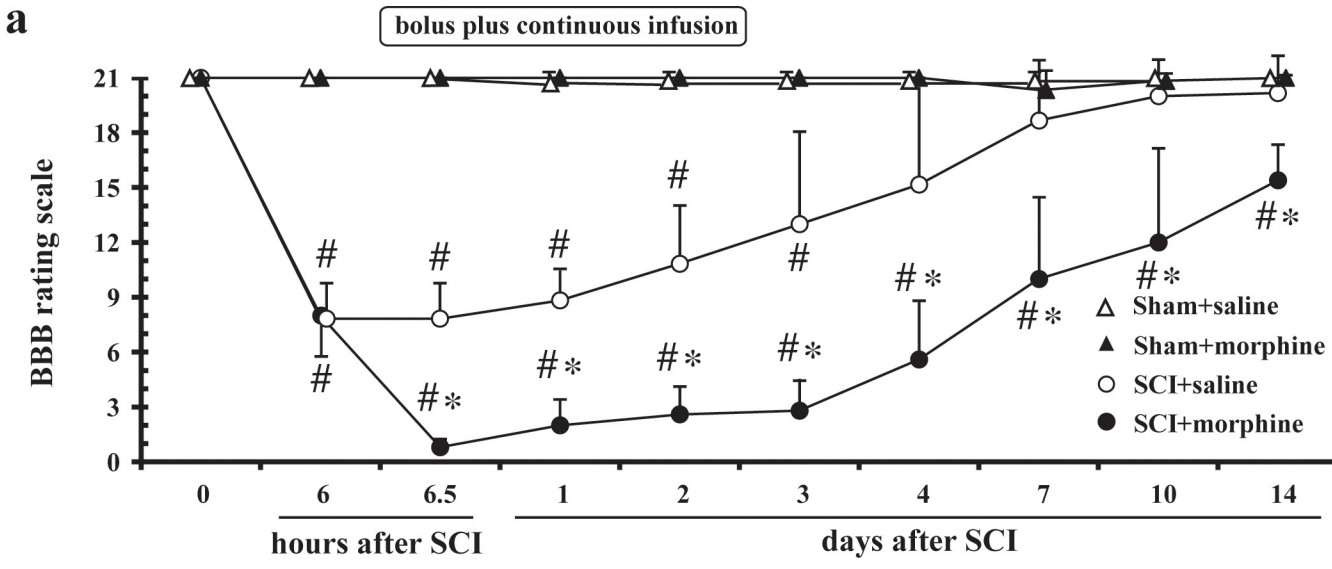

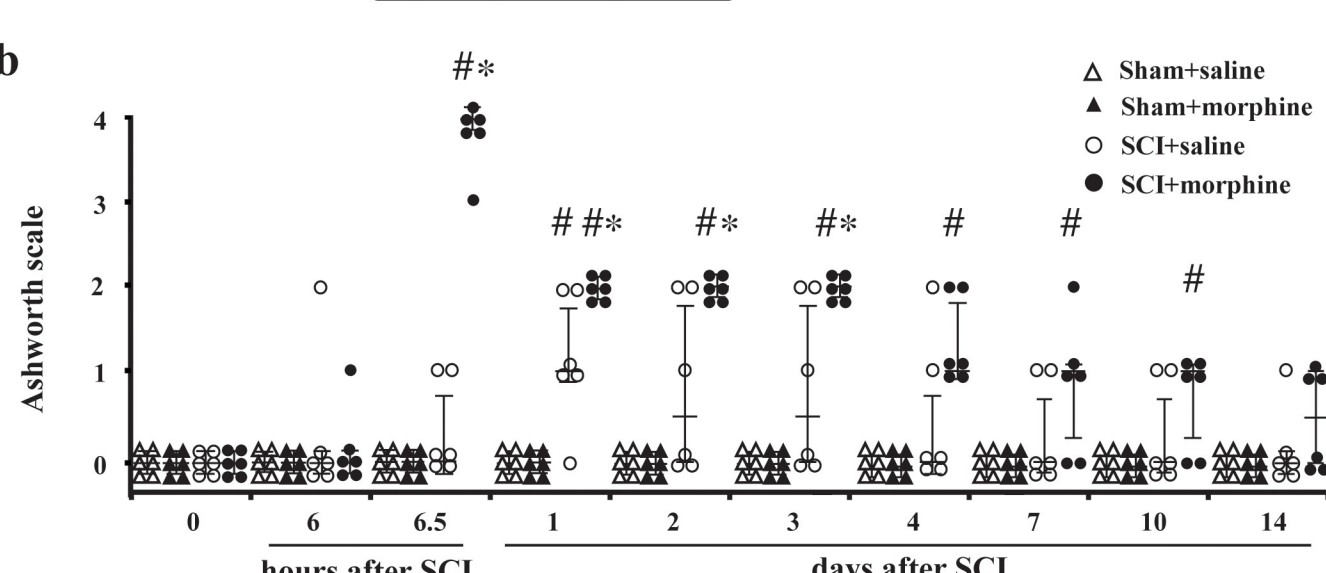

**Fig 3. Effects of continuous infusion of intrathecal (IT) morphine on locomotor function and muscle tone after mild spinal cord injury (SCI).**
Locomotor function (a) was evaluated using the 21-point Basso, Beattie, and Bresnahan (BBB) locomotor rating scale and muscle tone (b) was evaluated using the Ashworth scale before SCI (time 0) and 6 hours after the SCI or sham operation and after the start of continuous administration of IT morphine or normal saline up to day 14. A single dose of 30 μg of IT morphine was administered followed by continuous infusion of morphine (3 μg/hour with a concentration of 3 μg/1 μl of morphine in saline) for 72 hours to rats with SCI (SCI + morphine) and sham-operated rats without SCI (Sham + morphine). Physiological saline was continuously administered (1 μl/hour) for 72 hours to rats with SCI (SCI + saline) and sham-operated rats without SCI (Sham + saline). Data of the BBB scores are expressed as means ± SDs. n = 6 in each group. Data are presented as scatter dot plots of the Ashworth scores displaying the median as a line and the 25–75 percentiles. * $P < 0.05$ compared to SCI + saline group. # $P < 0.05$ compared to Sham + saline group.

analgesic doses of DAMGO, DPDPE and U50,488H for antinociception and antiallodynic action on neuropathic pain are 87 ng, 84 μg and 25 μg, respectively [19, 20], which are within the dose ranges used in this study. These analgesic doses of DAMGO, DPDPE and U50,488H did not produce motor impairment in animals without SCI (S1 Fig). It should be noted that

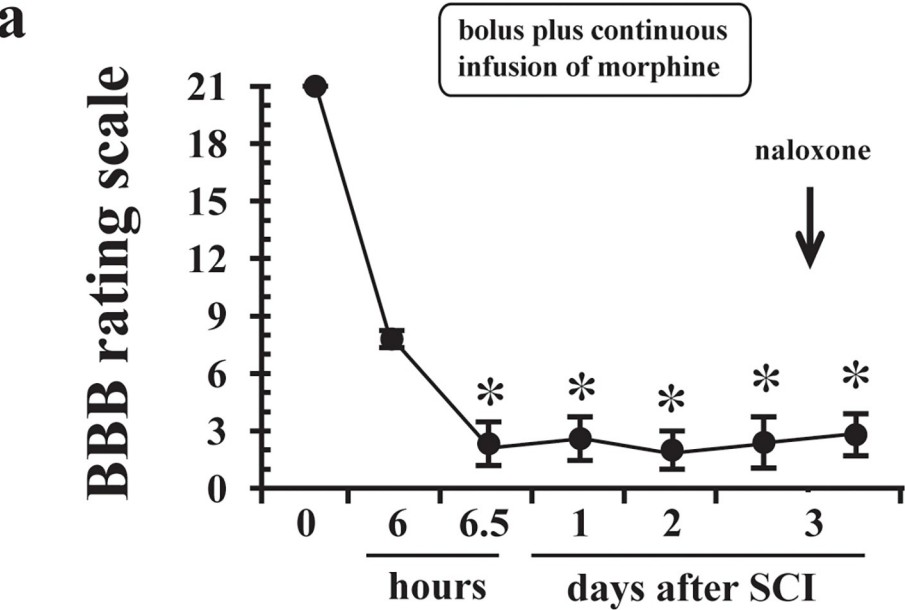

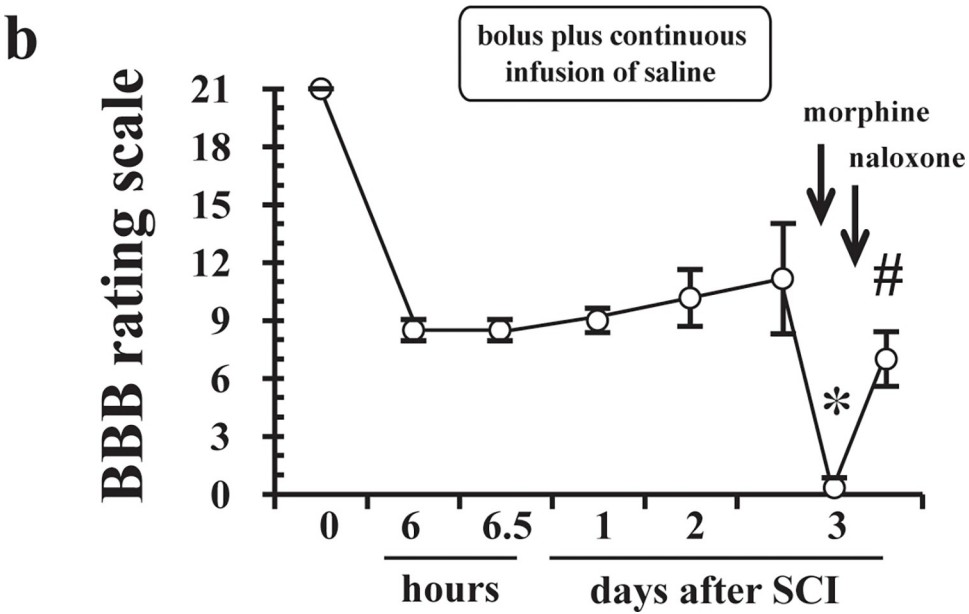

**Fig 4. Effects of intrathecal (IT) naloxone, an opioid receptor antagonist, on locomotor function after administration of IT morphine to rats with mild spinal cord injury (SCI).** Locomotor function was evaluated using the 21-point Basso, Beattie, and Bresnahan (BBB) locomotor rating scale. Panel (a) shows effects of IT naloxone (60 μg), administered 30 min after the end of 72-hour continuous infusion of morphine, on locomotor function of rats. A single dose of 30 μg of IT morphine was administered followed by continuous infusion of 3 μg/hour morphine for 72 hours to rats with SCI. Panel (b) shows effects of IT naloxone on locomotor function after IT administration of a single dose of morphine. A single dose of 30 μg of IT morphine was administered 30 min after the end of 72-hour continuous infusion of normal saline to rats with SCI. After another 30 min, naloxone was intrathecally administered. The BBB scores on the third day were evaluated 30 min after administration of each drug. Data are shown as means ± SDs. n = 6 in each group. $^{*}P < 0.05$ compared to data at 6 hours after thoracic SCI. # $P < 0.05$ compared to before IT naloxone.

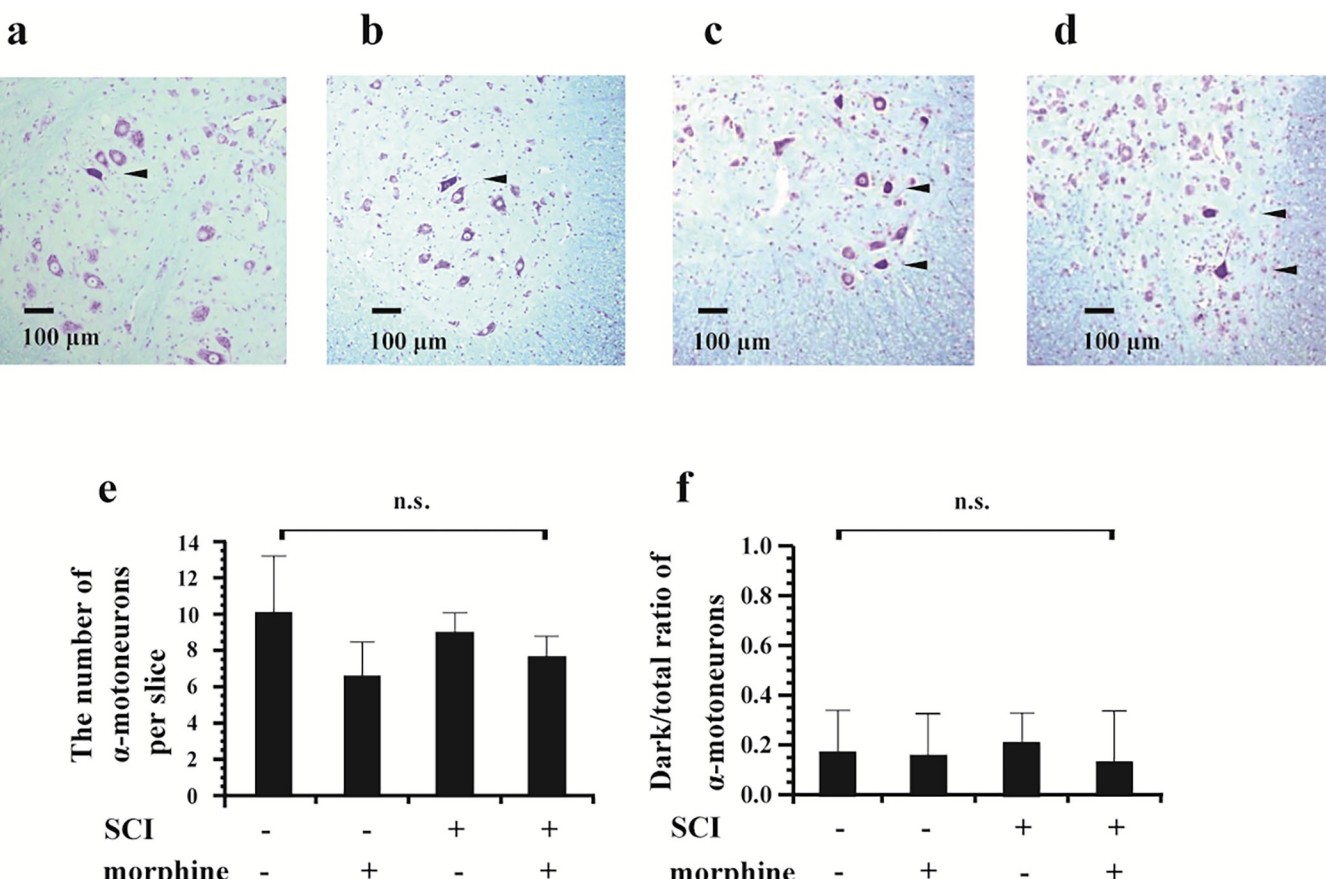

**Fig 5. Effects of continuous administration of intrathecal (IT) morphine on α-motoneurons in the lumbar spinal cord after mild thoracic spinal cord injury (SCI).** The ventral horns in the lumbar spinal cord were stained by the Klüver-Barrera method after 72-hour continuous infusion of IT normal saline to rats without SCI (a, sham + saline), IT morphine to rats without SCI (b, sham + morphine), IT normal saline to rats with mild thoracic SCI (c, SCI + saline), or morphine (d, SCI + morphine) to rats with mild thoracic SCI. Images of the left ventral horns are shown in this figure. The number of α-motoneurons in the left ventral horn of the lumbar spinal cord (e) was counted. Dark-stained α-motoneurons (considered to be damaged) (arrowhead) were seen in all of the groups and the ratios of damaged /total α-motoneurons (f) were calculated. Data are expressed as means ± SDs. n = 6 in each group.

DAMGO at a dose much lower than the analgesic dose increased muscle tone and aggravated residual locomotor function in the early phase of SCI in this study. Therefore, morphine-induced aggravation of locomotor function may be mainly mediated via activation of spinal μ-opioid receptors.

It has been shown that systemic [21–23], epidural [24] and IT opioids [25] at doses greatly exceeding analgesic doses induce muscular rigidity in normal humans and rats without SCI. SCI at the thoracic spinal cord level affects descending motor control and alters the balance of supraspinal excitatory and inhibitory inputs to α-motoneurons in the lumbar spinal cord. The responses of spinal neurons to inhibitory neurotransmitters such as glycine and gamma aminobutyric acid are inhibited by opiate alkaloids including morphine [26]. Alpha-motoneurons below the level of the lesion may become predisposed toward excitation because of decreases in inhibitory control [27, 28]. Thus, opioids at analgesic doses may increase muscle tone of the hindlimbs in the case of SCI.

A previous study showed little damage of motoneurons at a distance of 4 mm from the contusive epicenter [29]. Therefore, it is presumed that the effects of mild contusion injury at the T9-T10 spinal level did not extend to the lumbar spinal cord. This is also supported by our

results showing that mild thoracic SCI did not increase the number of damaged α-motoneu-rons in the lumbar spinal cord compared to sham surgery without SCI (Fig 5). As shown in Fig 5, approximately 15 percent of the α-motoneurons in the lumbar spinal cord were darkly stained (damaged) in all four groups with or without SCI and with or without morphine administration. It has been shown that intrathecal catheter placement can induce subclinical damage in the spinal cord and fascicles in contact with the catheter [30, 31]. Intrathecal cathe-ters were placed in rats in all four groups in the present study. The tip of the intrathecal cathe-ter was located at the lumbar enlargement of the spinal cord. Therefore, damage of some α-motoneurons observed in the four groups was more likely to be due to mechanical contact of the intrathecal catheter. Our results showed that continuous infusion of morphine did not increase the number of dark-stained α-motoneurons in the lumbar spinal cord. Taken together, the results suggest that the morphine-induced persistent decline in locomotor func-tion is not due to loss of α-motoneurons in the lumbar spinal cord.

Aggravated locomotor function after continuous infusion of morphine could not be reversed by IT naloxone (Fig 4A), in contrast to that after a single dose of 30 μg of morphine at both 6 hours (Fig 2A) and 72 hours (Fig 4B) after SCI. These results suggest that the mecha-nisms of persistent decline in locomotor function caused by continuous infusion of morphine are different from those of acute and transient decline of locomotor function caused by a single dose of morphine. We investigated the short-term effects of opioid receptor subtype-selective agonists on locomotor function (Fig 2), but we did not investigate their long-term effects. Con-tinuous infusion of naloxone would be required to completely block the activation of opioid receptors during continuous infusion of morphine because the half-life of naloxone is approxi-mately 10 min [32]. However, it was not tested in this study. Therefore, our results did not reveal in detail how activation of opioid receptors is involved in the persistent decline and delayed recovery of locomotor function in rats with mild SCI receiving continuous infusion of morphine for 72 hours. Antagonism of the κ-opioid receptor has been reported to attenuate the morphine-induced persistent decline of locomotor function by reducing the extent of cell death at the site of injury [33]. It has also been shown that an agonist of the κ-opioid receptor undermines the recovery of locomotor function after a moderate degree of SCI [34]. These results indicate that the κ-opioid receptor plays a critical role in the morphine-induced attenu-ation of locomotor recovery. It should be noted that there was a difference in the degrees of SCI in those previous studies [33, 34] and our study. Locomotor function at 1 day after SCI in the previous studies was lower than that in our study using rats with mild SCI. Morphine-induced acute deterioration of locomotor function after SCI, which was observed in our study, may occur only in the case of mild SCI in which locomotor function is preserved to some extent. The results of our study taken together with the results of those previous studies [33, 34] suggest that morphine has various impacts on residual locomotor function after mild SCI, including acute deterioration and attenuation of recovery, via different subtypes of opioid receptors.

The BBB score of rats receiving 72-hour continuous infusion of morphine at 14 days after SCI was significantly lower than that of rats receiving saline, although it was on a recovery trend (Fig 3A). Therefore, it is not clear from the present study whether continuous infusion of morphine simply delays functional recovery or leads to permanent decline in locomotor function. At present, it is not clear why the aggravated locomotor function after continuous infusion of morphine did not recover immediately after discontinuing morphine administra-tion. In our study, the number of the α-motoneurons in the lumbar spinal cord was evaluated. Neuronal sprouting in the spinal cord was not histologically investigated in the present study because sprouting of primary afferent fibers and the corticospinal tract does not occur imme-diately after SCI [35, 36]. Mechanisms other than involvement of activation of spinal opioid

receptors and effects of morphine on the injured tissue in the thoracic spinal cord could not be revealed from the present study. Further work will be necessary to clarify the mechanisms of morphine-induced deterioration in locomotor recovery from mild SCI.

Some limitations exist regarding the present study. First, while DPDPE and U50,488H used in this study agonize δ1- and κ1-opioid receptors, respectively [37], involvement of other subtypes such as δ2- and κ2-opioid receptors remains unknown from the present study. It has been reported that GR89696, a κ2-opioid receptor agonist, attenuates motor recovery at 3 weeks after SCI [34]. We evaluated the acute effects, but not long-term effects, of morphine and selective opioid receptors on motor function after SCI. Second, it has been reported that the expression of opioid receptors below the SCI changes within one to two days or several hours after SCI [38, 39]. In the present study, changes in the expression of opioid receptors after mild SCI was not investigated. Third, locomotor function recovery assessed by the BBB rating scale has been reported to be correlated with the spared tissues. However, rats can recover nearly normal locomotion even if some damage remains in the epicenter region [40]. Histological assessment in the lesion epicenter is required in a future study to investigate how or whether morphine affects the spinal cord after mild injury.

Our results suggest that a single dose of IT morphine at an analgesic dose in the early phase of mild mid-thoracic SCI aggravates residual locomotor function with increases in muscle tone of the hindlimbs via activation of spinal μ- and δ- opioid receptors. In addition, an increased dose of morphine with prolonged administration may stimulate mechanisms other than direct activation of classical opioid receptors, resulting in a sustained decline in locomotor function after SCI. Morphine should be used cautiously with close monitoring even in cases of mild SCI.

## Supporting information

**S1 File. The supporting information file includes data before statistical analysis.** (XLSX)

**S1 Fig.** Locomotor function (a-d) and muscle tone (e-h) after intrathecal (IT) administration of morphine and μ- (DAMGO), δ- (DPDPE), and κ- (U50,488H) selective opioid receptor agonists after sham surgery without spinal cord injury. Locomotor function was evaluated using the 21-point Basso, Beattie, and Bresnahan (BBB) locomotor rating scale (0 = complete paralysis, 21 = normal locomotion) before sham surgery (baseline), 6 hours after sham surgery (sham surgery), and 30 min after administration of each drug. The panels show effects of (a) morphine, (b) DAMGO, (c) DPDPE, and (d) U50,488H. Data of the BBB score are presented as mean ± SDs. The Ashworth scales (0 = no increase in tone; 4 = increase in muscle tone) were used to evaluate changes in muscle tone of the hindlimbs before sham surgery (baseline), 6 hours after sham surgery (sham surgery), and 30 min after administration of each drug. The panels show effects of (e) morphine, (f) DAMGO, (g) DPDPE, and (h) U50,488H. Data are presented as scatter dot plots of the Ashworth score displaying the median as a line and the 25–75 percentiles. n = 4 for each dose. (TIF)

**S2 Fig. Relationship between locomotor function and muscle tone.** Scatter plots showing the relationship between the BBB score (y-axis) and the Ashworth score (x-axis) 6 hours after mild SCI (yellow), 30 min after intrathecal administration of 30 μg of morphine (panel a), 10 ng of DAMGO (μ-opioid receptor agonist, panel b), 100 μg of DPDPE (δ-opioid receptor agonist, panel c), and 150 μg of U50,488H (κ-opioid receptor agonist, panel d) (red) and 30 min after intrathecal administration of naloxone (opioid receptor antagonist) (blue). Naloxone was

not used in the rats receiving U50,488H because the BBB scores did not decrease after administration of U50,488H. The plot shows the data for all rats (n = 6 in each group), with each rat indicated by a different shaped mark.

(TIF)

## Acknowledgments

We are grateful to Professor Jun Nakayama (Department of Pathology, Shinshu University School of Medicine, Matsumoto, Nagano, Japan) for his help in reviewing the pathology of the spinal cord.

## Author Contributions

**Conceptualization:** Mikito Kawamata.

**Data curation:** Katsuhiro Kawakami, Yuki Sugiyama.

**Funding acquisition:** Katsuhiro Kawakami.

**Investigation:** Katsuhiro Kawakami, Yuki Sugiyama, Noriaki Mochizuki.

**Methodology:** Satoshi Tanaka.

**Project administration:** Satoshi Tanaka.

**Supervision:** Mikito Kawamata.

**Writing – original draft:** Katsuhiro Kawakami.

**Writing – review & editing:** Satoshi Tanaka, Mikito Kawamata.

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
