## [Decision Letter · Decision Letter 0]

19 Nov 2021

PONE-D-21-29555Intrathecal morphine exacerbates paresis with increasing muscle tone of hindlimbs in rats with mild thoracic spinal cord injury but without damage of lumbar a-motoneuronsPLOS ONE

Dear Dr. Tanaka,

Thank you for submitting your manuscript to PLOS ONE. After careful consideration, we feel that it has merit but does not fully meet PLOS ONE’s publication criteria as it currently stands. Therefore, we invite you to submit a revised version of the manuscript that addresses the points raised during the review process.

ACADEMIC EDITOR:While the Reviewer found the work interesting, there are some issues that needs to be addressed.   Please review the comments and we encourage you to resubmit.

We look forward to receiving your revised manuscript.

Kind regards,

Warren Joseph Alilain, PhD

Academic Editor

PLOS ONE

Journal Requirements:

2. Please include the following information relating to animal experiments in your Methods section:

-The frequency of animal monitoring, including the specific criteria you used to monitor animal health;

-Animal welfare considerations, including efforts to alleviate suffering, such as:

-Postoperative care,

-The method of euthanasia

Reviewers' comments:

Reviewer's Responses to Questions

**Comments to the Author**

1. Is the manuscript technically sound, and do the data support the conclusions?

Reviewer #1: Partly

2. Has the statistical analysis been performed appropriately and rigorously? 

Reviewer #1: Yes

3. Have the authors made all data underlying the findings in their manuscript fully available?

Reviewer #1: No

4. Is the manuscript presented in an intelligible fashion and written in standard English?

Reviewer #1: Yes

5. Review Comments to the Author

Reviewer #1: One of the underlying goals of this work is to determine how a single dose of opioids influences lower extremity function acutely following a high thoracic injury where the HL MNs are presumably intact. A second goal is to determine how continuous delivery of opioids influences locomotor recovery over a slightly longer period. The key outcomes are the BBB scale for hindlimb function during stepping and the Ashworth scale that looks at hindlimb rigidity.

Overall, the study addresses important issues. Some of the data/results are of interest, however there are some fairly serious experimental design issues that render the data hard to interpret for both major components of the study.

Major issues:

The authors indicate multiple times that previous literature shows intrathecally delivered morphine to increase spinal cord injury lesion size, specifically around the epicenter of the injury site. The authors speculate that this phenomenon may contribute to the locomotor deficits that are seen in the study, however, no analysis was performed on the injury epicenter, spared white matter, or the extent of damage after SCI. This renders the results very hard to interpret, in particular in light of the issues mentioned below.

The study design does not allow the interpretation that 72h infusion of morphine induces a lasting functional difference, just a delay in recovery. The text should clearly reflect this.

The graphs in Figures 1, 3a and 4 represent the data as continuous without scaling the x-axis for time. This is misleading and the graphs need to be re-done to clearly show what is continuous and the temporal relationships in the data.

More specifically:

Results:

It is a shame that the experiment was not extended out to day 28, because it appears that the significant difference shown in Figure 4 would be gone by days 21 or 28. This suggests that whatever impact continuous morphine had on the circuitry is not going to induce a change in terminal function. The relationship between Ashworth and BBB data should be investigated using a scatterplot to look for correlations on an animal by animal basis.

MN counts. Please described if and how you used stereological principles and prevented double-counting MNs given that the sections were 5µm in diameter and MNs may appear in several consecutive sections. Also, more information is necessary for the reader to understand the criteria you used to identify both uninjured and injured or damaged MNs. Did you only count MNs with visible nucleoli? Did damaged MNs also have to be greater than 25µm in diameter?

The images in Figure 5 are confusing and the location of the MNs in a is quite different from in b and c. Are you certain the orientation of the sections are similar?

How long did you wait after introducing the next drug or drug concentration before assessing the BBB and ashworth? This is an important detail.

For the data shown in Figure 4, did you “wash out” morphine before delivering naloxone, and for how long?

Discussion:

On pages 18 and 19 (lines 318 to 322) you discuss damage to the MNs, but there are a couple issues of logic. First of all, the section shown in Figure 5a shouldn’t have any injured MNs because it is a control animal that received morphine only. Thus, the lack of increase following injury is in question. The second logic problem is the presumption that a mild contusion injury at T9-10 did not “extend to the lumbar spinal cord”. No doubt that the frank injury and cavitation did not, but even a mild injury disrupts descending and propriospinal input onto motoneurons and pre-motor neurons which could definitely influence their “health”. This should be re-stated and re-interpreted. This is particularly important because of the discussion on lines 335- about the descending input into the lumbar spinal cord.

Figures:

Figure 1. This figure is important and interesting, but shows each time point as having the same value over time. This is very misleading and mis-represents the data. I suggest showing this as two separate graphs, one to illustrate the Pre-SCI, Sci, 30min and day 1, and one to show the whole dataset (without the 30min), with the x-axis scaled to time. Or, keep the data all on one graph but scale the x-axis to time.

Same thing for Figure 3a. This is very misleading and in fact doesn’t even indicate the influence of the single dose of morphine given prior to the 30min assessment. This is absolutely critical and needs to be shown in some other way.

In Figure 2, although the and κ-opioid receptor agonist did not cause significant changes in function, Panels d and h should include the effects of reversing the κ-opioid receptor agonist with naloxone as Panels a-c/e-j showed for the other agonists and morphine.

In Figure 4a, the naloxone data is not collected on day 4 and thus the data points should not be shown connected. Same thing for figure 4b (morphine and naloxone).

6. PLOS authors have the option to publish the peer review history of their article (what does this mean?). If published, this will include your full peer review and any attached files.

Reviewer #1: No

---

## [Author Response · Author response to Decision Letter 0]

16 Jan 2022

We would like to thank the academic editor and reviewer for carefully reading out manuscript (PONE-D-21-29555) and for the insightful comments. The comments led us to an improvement of the work. Detailed responses to the academic editor and reviewer are shown in the file “Responses to Reviewers”. Thank you for giving us the opportunity to strengthen our manuscript with your valuable comments and queries.

---

## [Decision Letter · Decision Letter 1]

25 Apr 2022

PONE-D-21-29555R1

Intrathecal morphine exacerbates paresis with increasing muscle tone of hindlimbs in rats with mild thoracic spinal cord injury but without damage of lumbar a-motoneurons

PLOS ONE

Dear Dr. Tanaka,

Thank you for submitting your manuscript to PLOS ONE. After careful consideration, we feel that it has merit but does not fully meet PLOS ONE’s publication criteria as it currently stands. Therefore, we invite you to submit a revised version of the manuscript that addresses the points raised during the review process.

While one reviewer has acknowledged some revisions and offers more, they point out that there remain shortcomings mentioned in the previous reviews that should be addressed to strengthen the report. Another reviewer has posed further pertinent queries that must be addressed equally, in particular discussion of current literature that differs from current contentions regarding morphine treatment after SCI in rodents. 

We look forward to receiving your revised manuscript.

Kind regards,

Alexander Rabchevsky, Ph.D.

Academic Editor

PLOS ONE

Reviewers' comments:

Reviewer's Responses to Questions

**Comments to the Author**

1. If the authors have adequately addressed your comments raised in a previous round of review and you feel that this manuscript is now acceptable for publication, you may indicate that here to bypass the “Comments to the Author” section, enter your conflict of interest statement in the “Confidential to Editor” section, and submit your "Accept" recommendation.

Reviewer #1: (No Response)

Reviewer #2: (No Response)

2. Is the manuscript technically sound, and do the data support the conclusions?

Reviewer #1: Partly

Reviewer #2: Yes

3. Has the statistical analysis been performed appropriately and rigorously? 

Reviewer #1: Yes

Reviewer #2: Yes

4. Have the authors made all data underlying the findings in their manuscript fully available?

Reviewer #1: Yes

Reviewer #2: Yes

5. Is the manuscript presented in an intelligible fashion and written in standard English?

Reviewer #1: Yes

Reviewer #2: Yes

6. Review Comments to the Author

Reviewer #1: While the authors have been reasonably responsive to the reviews, and the manuscript is much improved, there remain some shortcomings mentioned in the previous review that, if addressed, would greatly strengthen the interpretation.

1. The authors speculate in the discussion section that motor neurons below the level of the lesion may receive more excitatory input after SCI, contributing to the observed changes in muscle tone. However, they don’t provide any evidence for this, even though it would be quite easy. IHC for CGRP and analysis of afferent sprouting would be very revealing.

2. An additional very easy assessment, assessment of descending, supraspinal input onto lumbar motor neurons (IHC for 5-HT for example) would be very helpful.

3. An analysis of opioid receptor populations and their distributions.

4. The motor neuron viability/damage analysis is problematic, in particular in light of the authors contention that mechanical injury might occur during pump implantation. These groups should be compared to normal non-implanted animals (with and without administration of morphine) to help strengthen the motor neuron viability component of the paper.

Minor suggestions/comments:

Page 8. It might be better to say that “Each rat’s bladder was manually expressed twice daily”, since this action is a little more subtle than just squeezing.

Page 9. 5 rats met ethical endpoints. Which groups were they in? If morphine influenced the expression of autophagy or lethargy then this should be explained and hopefully explored?

For the data shown in Figure 1, it is important that the reader knows that the morphine was given after the 6hr BBB assessment, and that you acknowledge the slight difference in group BBB at that time (before morphine, correct?). This group difference persists even if it is not statistically significant, and may have nothing to do with morphine. Please add the detail that morphine was given after the 6hr BBB assessment (to the legend or to the text in the results section).

Related to #4 above, the concept that MNs densely stained with nissl are damaged or dying should be supported by a reference (or references) in the methods, results and discussion.

Reviewer #2: This study adds to the growing evidence of adverse effects of morphine on the prognosis for recovery after SCI. Using a mild SCI model, the data demonstrates that continuous administration of morphine, for 3 days, undermines recovery. The study also documents the spastic hypertonia that is noted with acute morphine administration, while it is active. While many of the findings are not particularly novel, the paper is important, increasing the data on effects of early opioid administration after SCI, a highly significant clinical concern.

For the methods, please clarify where the tip of the catheter lay in relation to the spinal injury, was it caudal or centered over the lesion site? What experimental groups were the 5 rats that were euthanized before 14 days in? Was there a bias with increased autophagia noted in morphine-treated subjects, as has been seen in other studies? With the implanted osmotic pump, how did you verify that it was active and released sufficient morphine over the 14 days? What is the half-life of DPDPE, DAMGO and U50488H, relative to morphine?

Withdrawal from continuous morphine administration, with or without naloxone did not improve recovery. It would have been interesting to know whether early naloxone administration blocked the effects of continuous morphine (if given continuously also). Are there any previous studies addressing this? This should be included in the discussion.

I am not 100% sure of the point of the short-term tests of each of the opioids on locomotor recovery. To some extent it seems logical that the rats may reduce locomotion, and appear less coordinated, while opioids are actively inhibiting activity in the spinal cord. A statement regarding the hypothesis being tested, or the implications of these studies, would be helpful, either in the methods or as part of the discussion.

In some ways, the results of this study also differ significantly from previously published reports (see papers by Faden and Aceves) that implicate kappa opioid receptor activation in reducing recovery after SCI. The authors should acknowledge and discuss these differences, with any explanation as to why the studies may be different. For example, are there differences in the mechanistic action of U50488H, GR89696, and dynorphin that could explain the disparate results?

7. PLOS authors have the option to publish the peer review history of their article (what does this mean?). If published, this will include your full peer review and any attached files.

Reviewer #1: No

Reviewer #2: No

---

## [Author Response · Author response to Decision Letter 1]

3 Jun 2022

We would like to thank the academic editor and reviewers for carefully reading our manuscript and for the insightful comments. The comments led to an improvement of the work. Detailed responses to the reviewers are shown below. The comments from the reviewers are shown in black and our replies are shown in red. In this revised version, changes to our manuscript within the document were all highlighted by using red-colored text (with underlining) in the file named “Revised Manuscript with Track Changes”.

Reply to Reviewer #1

Reviewer #1: While the authors have been reasonably responsive to the reviews, and the manuscript is much improved, there remain some shortcomings mentioned in the previous review that, if addressed, would greatly strengthen the interpretation.

1. The authors speculate in the discussion section that motor neurons below the level of the lesion may receive more excitatory input after SCI, contributing to the observed changes in muscle tone. However, they don’t provide any evidence for this, even though it would be quite easy. IHC for CGRP and analysis of afferent sprouting would be very revealing.

We greatly appreciate the useful suggestions. A number of studies have shown that CGRP fibers become more prevalent both rostral and caudal to the injured spinal cord, and this has been generally interpreted as sprouting of the fibers (Krenz and Weaver, Neuroscience 1998; Ondarza et al., Exp Neurol 2003). CGRP immunostaining studies would provide useful information on regeneration of sensory fibers. The area of CGRP-immunoreactive fibers was increased in all cord segments at 2 weeks but not at 1 week after spinal cord transection (Krenz and Weaver). It is presumed that the impact of sensory fiber regeneration on motor function is small at 6 hours and 3 days after spinal cord injury (SCI), when morphine was administered in our study. Therefore, we did not investigate regeneration of the sensory fibers after administration of morphine to rats with thoracic SCI.

Nonetheless, the reviewer’s suggestions are important, and we have included your point as a consideration for future study. The following description (underlined) has been added to the Discussion section in the revised manuscript, in line 394.

“The lesion epicenter in the thoracic spinal cord was not histologically investigated. Neuronal sprouting in the spinal cord was not histologically investigated in the present study because sprouting of primary afferent fibers and the corticospinal tract does not occur immediately after SCI (Krenz and Weaver, Neuroscience 1998; Donnelly and Popovich, Exp Neurol 2008). Mechanisms other than involvement of activation of spinal opioid receptors could not be revealed from the present study. Further work will be necessary to clarify the mechanisms of morphine-induced deterioration in locomotor recovery from mild SCI.”

2. An additional very easy assessment, assessment of descending, supraspinal input onto lumbar motor neurons (IHC for 5-HT for example) would be very helpful.

As the reviewer pointed out, an immunohistochemical study for 5-HT could be a useful method for assessing the disruption and regeneration of descending serotonergic projections to the spinal motor areas. Corticospinal tract (CST) sprouting occurs between 3 weeks and 3 months after contusive SCI (Hill et al., Exp Neurol 2001). Therefore, it is presumed that the impact of CST sprouting on the spinal motor areas is small at 6 hours and 3 days after spinal cord injury (SCI), when morphine was administered in our study. Therefore, damage of descending fibers in the spinal cord was not investigate in this study.

3. An analysis of opioid receptor populations and their distributions.

Thank you for your important suggestion. 

The proportions of the three main types of opioid binding sites in the spinal cord are considered to be as follows: 70.4-74.3%, 18.4-20.3% and 7.3-9.5% for μ, δ, and κ sites, respectively (Besse et al., Brain Res 1991). After SCI, the number of μ opioid receptors decreases (Michael FM et al., Neural Res 2015), the number of δ opioid receptors remains unchanged, and the number of κ opioid receptors increases (Krumins SA and Faden AI. Ann Neurol 1986). However, we did not investigate changes in opioid receptor expression after SCI. 

In line 399, we have added the following sentences (underlined) in the Discussion section.

“Some limitations exist regarding the present study. First, while DPDPE and U50,488H used in this study agonize δ1- and κ1-opioid receptors, respectively (Vankova et al., Anesthesiology 1996), involvement of other subtypes such as δ2- and κ2-opioid receptors remains unknown from the present study. Second, it has been reported that the expression of opioid receptors below the SCI changes within one or two days after SCI (Michael FM, et al., Neurol Res. 2015; Krumins SA and Faden AI. Ann Neurol 1986). In the present study, changes in the expression of opioid receptors after mild SCI was not investigated. Third, locomotor function recovery assessed by the BBB rating scale has been reported to be correlated with the spared tissues. However, rats can recover nearly normal locomotion even if some damage remains in the epicenter region (Basso DM, et al, Exp Neurol 1996). Histological assessment in the lesion epicenter is required in a future study to investigate how or whether morphine affects the spinal cord after mild injury”

4. The motor neuron viability/damage analysis is problematic, in particular in light of the authors contention that mechanical injury might occur during pump implantation. These groups should be compared to normal non-implanted animals (with and without administration of morphine) to help strengthen the motor neuron viability component of the paper.

In the present study, �-motoneurons in the lumbar spinal cord of non-catheterized rats were not histologically evaluated. In this revised version of the manuscript, we added data showing �-motoneurons of catheterized rats without thoracic SCI and without administration of morphine. The number of dark-stained �-motoneurons in this group was similar to that in other groups (no SCI+ morphine, SCI + saline, and SCI + morphine). The tip of the catheter was located in the lumbar spinal cord. Therefore, it is presumed that the insertion of the catheter may have damaged a small number of �-motoneurons to the extent that it did not lead to locomotor dysfunction. It is important to emphasize from figure 5 that continuous infusion of IT morphine did not result in increased loss of �-motoneurons in the lumbar spinal cord of rats with thoracic SCI. 

In this revised version, we changed in line 368 as follows (underlined). 

“As shown in Figure 5, approximately 15 percent of the �-motoneurons in the lumbar spinal cord were darkly stained (damaged) in all four groups with or without SCI and with or without morphine administration. It has been shown that intrathecal catheter placement can induce subclinical damage in the spinal cord and fascicles in contact with the catheter (Yaksh TL et al., Anesthesiology 1986; Sakura S et al., Anesthesiology 1996). Intrathecal catheters were placed in rats in all four groups in the present study. The tip of the intrathecal catheter was located at the lumbar enlargement of the spinal cord. Therefore, damage of some �-motoneurons observed in the three four groups is was more likely to be due to mechanical contact of the intrathecal catheter. than the thoracic SCI or morphine. Implantation of the intrathecal catheter but not. Our results showed that continuous infusion of morphine did not increase the number of darked-stained �-motoneurons in the lumbar spinal cord. continuous infusion of IT morphine per se would cause some loss of motoneurons in the lumbar spinal cord of rats with thoracic SCI.” 

Minor suggestions/comments:

Page 8. It might be better to say that “Each rat’s bladder was manually expressed twice daily”, since this action is a little more subtle than just squeezing.

Thank you for the advice. According to the reviewer’s suggestion, we have changed as follows in line 114 “Each rat’s bladder was manually expressed twice daily.”

Page 9. 5 rats met ethical endpoints. Which groups were they in? If morphine influenced the expression of autophagy or lethargy then this should be explained and hopefully explored?

All five animals that met humane endpoints received continuous infusion of morphine after thoracic SCI (SCI+MOR group). Three of the rats were lethargic and two displayed self-mutilation. Autophagia is thought to result from neuropathic pain in animals with lesions in the central nervous system (Mailis et al., Pain 1996; Frost et al., J Spinal Cord Med 2008). It has been reported that intrathecal morphine treatment did appear to enhance autophagia in animals with SCI, suggesting that morphine produces symptoms of neuropathic pain (Hook et al., J Neurotrauma, 2009). 

Aceves et al. reported that body weight in animals decreases in the acute phase of SCI (Aceves et al., J Neurotrauma 2017). In addition, Hook et al. reported that animals that received morphine exhibited greater weight loss than that in controls (Hook et al., J Neurotrauma 2009). Thus, in addition to SCI, morphine administration may lead to deterioration of the general condition.

In line 119, we added the following sentences. 

Five rats that received continuous infusion of morphine after thoracic SCI met the humane endpoints. Three of the rats were lethargic and two displayed self-mutilation. It has been reported that morphine administration leads to an increase in the incidence of autophagia and weight loss in animals with SCI compared to that without morphine (Hook et al., J Neurotrauma 2009). 

For the data shown in Figure 1, it is important that the reader knows that the morphine was given after the 6hr BBB assessment, and that you acknowledge the slight difference in group BBB at that time (before morphine, correct?). This group difference persists even if it is not statistically significant, and may have nothing to do with morphine. Please add the detail that morphine was given after the 6hr BBB assessment (to the legend or to the text in the results section).

As the reviewer pointed out, figure 1 may be difficult to understand.

In line 213, we have added the following sentences in the Results section.

“SCI at the T9-T10 spine level led to locomotor dysfunction but not complete paralysis of the hindlimbs. There were no significant differences in the BBB scores between the groups at 6 h after SCI before morphine administration (P = 0.085). The BBB score of the morphine group at 30 min after administration (6.5 hours after SCI) was significantly lower than that of the saline group (P = 0.006).” 

In addition, 

“An arrow indicates the timing of administration of drugs (morphine or saline)” was added to line 225 in the legend of figure 1.

Related to #4 above, the concept that MNs densely stained with nissl are damaged or dying should be supported by a reference (or references) in the methods, results and discussion.

Klüver-Barrera staining, which enables detection of abnormalities in the nucleus and myelin sheath at low magnification, has been used to evaluate motoneurons in the spinal cord. As in previous studies, darkly-stained α-motoneurons localized in the ventral horn were counted as impaired neurons (Cousins et al., Anesth Analg 2003).

We added the reference in the Methods section.

“Transversal sections of 5 �m in thickness were obtained by cutting the spinal cord at the level of the lumbar enlargement and were stained by the Klüver-Barrera method to assess neuronal damage (Cousins et al., Anesth Analg 2003).”

Reply to Reviewer #2

Reviewer #2: This study adds to the growing evidence of adverse effects of morphine on the prognosis for recovery after SCI. Using a mild SCI model, the data demonstrates that continuous administration of morphine, for 3 days, undermines recovery. The study also documents the spastic hypertonia that is noted with acute morphine administration, while it is active. While many of the findings are not particularly novel, the paper is important, increasing the data on effects of early opioid administration after SCI, a highly significant clinical concern.

Thank you very much for your comments.

For the methods, please clarify where the tip of the catheter lay in relation to the spinal injury, was it caudal or centered over the lesion site? 

In lines 95-97 in the previous version of the manuscript, it was stated that “A PE-10 polyethylene catheter (Becton Dickinson Primary Care Diagnostics, Sparks, MD, USA) was inserted 15 mm cephalad into the lumbar subarachnoid space at the L4-L5 intervertebral space.”. In other words, the tip of the intrathecal catheter was located at the lumbar enlargement of the spinal cord below the level of injury. Therefore, it is unlikely that SCI at T9-T10 was affected by the mechanical contact of the intrathecal catheter.

In line 373, we have added the following description. 

“The tip of the intrathecal catheter was located at the lumbar enlargement of the spinal cord.” 

What experimental groups were the 5 rats that were euthanized before 14 days in? Was there a bias with increased autophagia noted in morphine-treated subjects, as has been seen in other studies?　

As we responded to another reviewer, all five animals that met humane endpoints received continuous infusion of morphine after thoracic SCI (SCI+MOR group). Three of the rats were lethargic and two displayed self-mutilation. Autophagia is thought to result from neuropathic pain in animals with lesions in the central nervous system (Mailis et al., Pain 1996; Frost et al., J Spinal Cord Med 2008). It has been reported that intrathecal morphine treatment did appear to enhance autophagia in animals with SCI (Hook et al., J Neurotrauma, 2009). 

With the implanted osmotic pump, how did you verify that it was active and released sufficient morphine over the 14 days?

The osmotic pump (ALZET model 1003D) used in this study delivers solutions continuously at a rate of 1 �l/h. A pump filled with 72 �l of morphine or saline was implanted. Under sevoflurane anesthesia, the pump was removed at 14 days after the start of continuous infusion. Then it was confirmed that there was no residual solution in the pump. Because the pump seemed to be functioning without any problems, it was presumed that morphine or saline has been continuously infused for 3 days (72 hours).

In line 187, we have added the following sentences in the Methods section. 

“Under sevoflurane anesthesia, the pump was removed at 14 days after the start of continuous infusion. Then it was confirmed that there was no residual solution in the pump.”

What is the half-life of DPDPE, DAMGO and U50488H, relative to morphine?

When given intrathecally in mice, the half-lives of morphine, DAMGO, DPDPE are between 10 and 15 min (Heyman et al., Life Sci 1986). We could not find any report showing the half-life of U50,448H. Kakinohana et al. reported that the maximal effects of intrathecal morphine, DAMGO, DPDPE and U50,448H on the residual locomotor function after ischemic SCI appeared in 30 – 60 min (Kakinohana et al., Br J Anaesth 2006). Based on these results, we evaluated locomotor function and muscle tone 30 min after administration of the drugs (Figure 2 and 3).

Withdrawal from continuous morphine administration, with or without naloxone did not improve recovery. It would have been interesting to know whether early naloxone administration blocked the effects of continuous morphine (if given continuously also). Are there any previous studies addressing this? This should be included in the discussion.

That is an interesting point. In the present study, continuous infusion of morphine-induced decreases in locomotor function assessed by the BBB scale were not reversed by naloxone administered at 72 hours after SCI. However, to the best of our knowledge, there has been no report about the effects of naloxone given in the early stage of continuous infusion of morphine on locomotor function and muscle tone after SCI. 

The half-life of naloxone is approximately 10 min (Heyman et al., Life Sci 1986). Therefore, it is presumed that a single shot of naloxone at the beginning of continuous infusion of morphine does not improve subsequent recovery during continuous infusion of morphine. The effect of continuous administration of naloxone during continuous infusion of morphine was not investigated in this study. Further research is needed on these matters.

In line 385, we have the added following sentences.

“The half-life of naloxone is approximately 10 min (Heyman et al., Life Sci 1986). Therefore, it is presumed that subsequent locomotor recovery in rats with continuous infusion of morphine was not improved even if a single shot of naloxone was administered at the beginning of continuous infusion of morphine. The effect of continuous administration of naloxone during continuous infusion of morphine was not investigated in this study.”

I am not 100% sure of the point of the short-term tests of each of the opioids on locomotor recovery. To some extent it seems logical that the rats may reduce locomotion, and appear less coordinated, while opioids are actively inhibiting activity in the spinal cord. A statement regarding the hypothesis being tested, or the implications of these studies, would be helpful, either in the methods or as part of the discussion.

The present study showed that a single dose of IT morphine (30 �g) did not affect recovery but deteriorated locomotor function in the short term (Figure 1). The half-life of morphine is between 10 and 15 min (Heyman et al., Life Sci 1986). In contrast, continuous administration of morphine for 72 hours delayed functional recovery from SCI (Figure 3). It has been reported that a single dose of 90 �g morphine delays locomotor functional recovery after moderate SCI (Hook et al., J Neurotrauma 2009). Therefore, locomotor functional recovery from SCI may vary depending on the dose of morphine and duration of morphine administration. 

The data in Figure 2 were obtained to investigate which opioid receptors are involved in this short-term deterioration of locomotor function. The deterioration in locomotor function by DAMGO and DPDPE was probably due to an increase in muscle tone. The short-term morphine-induced locomotor dysfunction seen in Figure 1 was thought to be mediated by �- or δ-opioid receptors, because the effects were reversed by naloxone (Figure 2). However, decreases in locomotor function after continuous infusion of morphine were not reversed by naloxone.

The important findings of this study were that a single dose of morphine after mild thoracic SCI rapidly aggravated locomotor function with increases in muscle tone and that continuous infusion of morphine delayed recovery. 

In line 341, we have added the following sentences.

“It has been reported that a single dose of 90 �g morphine delays locomotor functional recovery after moderate SCI (Hook et al., J Neurotrauma 2009). Therefore, locomotor functional recovery from SCI may vary depending on the dose of morphine and duration of morphine administration.”

In some ways, the results of this study also differ significantly from previously published reports (see papers by Faden and Aceves) that implicate kappa opioid receptor activation in reducing recovery after SCI. The authors should acknowledge and discuss these differences, with any explanation as to why the studies may be different. For example, are there differences in the mechanistic action of U50488H, GR89696, and dynorphin that could explain the disparate results?

Thank you for your important suggestion. Faden et al. reported that WIN44,441-3, a kappa opioid receptor antagonist, promoted motor recovery at 4 weeks after SCI (Faden et al., Peptides 1985). Aceves et al. showed that nor-Binaltorphimine, a kappa opioid receptor antagonist, reversed morphine (single dose)-induced attenuation of locomotor recovery at 21 days after SCI (J Neurotrauma 2017). These results suggest that activation of kappa-opioid receptors at the acute phase of SCI is involved in deterioration of recovery at 3 to 4 weeks after SCI.

 In contrast to the studies described above, we administered a single dose of naloxone, a pan-opioid receptor antagonist, after continuous infusion of morphine for 72 hours (Figure 4). In other words, naloxone was administered later in the present study than in the previous studies in which the roles of kappa-opioid receptors were investigated. In addition, the duration of morphine administration was also longer in the present study than in the previous studies.

Our results suggest that the acute exacerbation of motor function following morphine-induced thoracic spinal injury is at least partially due to hypertonia of hindlimb muscles through activation of μ-opioid and δ-opioid receptors but not activation of κ-opioid receptors (Figure 2).

In contrast, as shown in Figure 4, naloxone did not reverse the locomotor dysfunction after 3-day continuous infusion of morphine, but it reversed the dysfunction caused by a single dose of morphine following 3-day continuous infusion of saline. This indicated that an increased dose of morphine with extended duration of administration affects motor function via mechanisms other than direct activation of opioid receptors. 

In line 385, the following sentences were added to the Discussion section.

 “Our results indicate that morphine-induced acute aggravation of locomotor function after thoracic SCI is due to hypertonia of the hindlimb muscles via activation of μ- and/or δ- opioid receptors but not κ-opioid receptors (Fig. 2). We did not investigate the impact of selective opioid receptor antagonists on the motor recovery after SCI because naloxone did not reverse the locomotor dysfunction after 3-day continuous morphine infusion (Fig. 4). It has been reported that activation of �-opioid receptors at the acute phase of SCI is involved in deterioration of recovery at 3 to 4 weeks after SCI (Faden et al., Peptides 1985; Aceves et al., J Neurotrauma 2017). The role of opioid receptors in motor function and recovery after SCI may change from time to time and differ by subtypes of opioid receptors”. 

For example, are there differences in the mechanistic action of U50488H, GR89696, and dynorphin that could explain the disparate results?

U50,488H is an agonist of kappa 1, GR 89696 is an agonist of kappa 2, and dynorphin is an agonist of kappa 1 and kappa 2. Aceves et al. showed (Spinal cord 2016) that GR 89696 (kappa 2 agonist) worsened recovery after SCI. These results indicate that the kappa 2 receptor is involved in the recovery of gait function after SCI. We used U50,448H, a kappa 1 agonist, to examine the acute effects of opioid receptor activation on attenuated locomotor function after SCI (Figure 2).

 In the Limitations section of the previous version of the manuscript, we already described as follows: “First, while DPDPE and U50,488H used in this study agonize δ1- and κ1-opioid receptors, respectively (Vankova et al., Anesthesiology 1996), involvement of other subtypes such as δ2- and κ2-opioid receptors remains unknown from the present study” 

"In line 399, we have added the following sentences.

” It has been reported that GR 89696, a κ2-opioid receptor agonist, attenuates motor recovery at 3 weeks after SCI. We evaluated the acute effects, but not the long-term effects, of morphine and selective opioid receptors on motor function after SCI.”

To make it easier for the reader to understand the manuscript, we have changed

or added the following sentences. (Add, Remove)

In line 176, from “into three groups,” to “into four groups,”

In line 178, from “3) sham-operated rats without SCI that received continuous IT morphine,” to “3) sham-operated rats without SCI that received continuous IT morphine, and 4) sham-operated rats without SCI that received continuous IT physiological saline.”

In line 179, from “In these 3 groups,” to “In these 4 groups,”

In line 280, from “Physiological saline was continuously administered (1 �l/hour) for 72 hours to rats with SCI (SCI + saline) “ to “Physiological saline was continuously administered (1 �l/hour) for 72 hours to rats with SCI (SCI + saline) and sham-operated rats without SCI (Sham + saline).” 

In line 284, from “sham + morphine group” to “sham + saline group”

In line 316, from “The ventral horns in the lumbar spinal cord were stained by the Klüver-Barrera method after 72-hour continuous infusion of IT morphine to rats without SCI (a, sham + morphine), IT normal saline to rats with mild thoracic SCI (b, SCI + saline), or morphine (c, SCI + morphine) to rats with mild thoracic SCI. Images of the left ventral horns are shown in this figure. The number of α-motoneurons in the left ventral horn of the lumbar spinal cord (d) was counted. Dark-stained α-motoneurons (considered to be damaged) (arrowhead) were seen in all of the groups and the ratios of damaged /total α-motoneurons (e) were calculated.” To “The ventral horns in the lumbar spinal cord were stained by the Klüver-Barrera method after 72-hour continuous infusion of IT normal saline to rat without SCI (a, sham + saline), IT morphine (b, sham + morphine) to rats without SCI, IT normal saline to rats with mild thoracic SCI (c, SCI + saline), or morphine (d, SCI + morphine) to rats with mild thoracic SCI. Images of the left ventral horns are shown in this figure. The number of α-motoneurons in the left ventral horn of the lumbar spinal cord (e) was counted. Dark-stained α-motoneurons (considered to be damaged) (arrowhead) were seen in all of the groups and the ratios of damaged /total α-motoneurons (f) were calculated.

In line 329, from “a single dose of morphine” to “a single dose of morphine (30 �g)“

We have added the following references and adjusted the serial number of other citations.

15. Cousins MJ, Kakinohana M, Fuchigami T, Nakamura S, Sasara T, Kawabata T, et al.,. Intrathecal administration of morphine, but not small dose, induced spastic paraparesis after a noninjurious interval of aortic occlusion in rats. Anesth Analg. 2003 ;96(3): 769-775. https://doi.org/10.1213/01.ANE.0000048855.24190.5F PMID: 12598261

30. Yaksh TL, Noueihed RY, Durant PA. Studies of the pharmacology and pathology of intrathecally administered 4-anilinopiperidine analogues and morphine in the rat and cat. Anesthesiology. 1986; 64(1): 54-66. https://doi.org/10.1097/00000542-198601000-00009 PMID: 2867722

31. Sakura S, Hashimoto K, Bollen AW, Ciriales R, Drasner K. Intrathecal catheterization in the rat. Improved technique for morphologic analysis of drug-induced injury. Anesthesiology. 1996; 85(5): 1184-1189. https://doi.org/10.1097/00000542-199611000-00028 PMID: 8916837

32. Heyman JS, Koslo RJ, Mosberg HI, Tallarida RJ, Porreca F. Estimation of the affinity of naloxone at supraspinal and spinal opioid receptors in vivo: studies with receptor selective agonists. Life Sci. 1986 ;39(19): 1795-1803. https://doi.org/10.1016/0024-3205(86)90099-8 PMID: 3022095

33. Faden AI, Knoblach S, Mays C, Jacobs TP. Motor dysfunction after spinal cord injury is mediated by opiate receptors. Peptides. 1985 ;6 Suppl 1: 15-17. https://doi.org/10.1016/0196-9781(85)90006-3 PMID: 2995941 

34. Aceves M, Bancroft EA, Aceves AR, Hook MA. Nor-Binaltorphimine Blocks the Adverse Effects of Morphine after Spinal Cord Injury. J Neurotrauma. 2017 ;34(6): 1164-1174. https://doi.org/10.1089/neu.2016.4601. PMID: 27736318

35. Krenz NR, Weaver LC. Sprouting of primary afferent fibers after spinal cord transection in the rat. Neuroscience. 1998 ;85(2):443-458. https://doi.org/10.1016/s0306-4522(97)00622-2. PMID: 9622243 

36. Donnelly DJ, Popovich PG. Inflammation and its role in neuroprotection, axonal regeneration and functional recovery after spinal cord injury. Exp Neurol. 2008 ;209(2):378-388. https://doi.org/10.1016/j.expneurol.2007.06.009. PMID: 17662717 

38. Michael FM, Mohapatra AN, Venkitasamy L, Chandrasekar K, Seldon T, Venkatachalam S. Contusive spinal cord injury up regulates mu-opioid receptor (mor) gene expression in the brain and down regulates its expression in the spinal cord: possible implications in spinal cord injury research. Neurol Res. 2015 ;37(9): 788-796. https://doi.org/10.1179/1743132815Y.0000000057 PMID: 26039701

39. Krumins SA, Faden AI. Traumatic injury alters opiate receptor binding in rat spinal cord. Ann Neurol. 1986; 19(5): 498-501. https://doi.org/10.1002/ana.410190514 PMID: 3013077

Again, thank you for giving us the opportunity to strengthen our manuscript with your valuable comments and queries.

---

## [Decision Letter · Decision Letter 2]

28 Jun 2022

PONE-D-21-29555R2Intrathecal morphine exacerbates paresis with increasing muscle tone of hindlimbs in rats with mild thoracic spinal cord injury but without damage of lumbar a-motoneuronsPLOS ONE

Dear Dr. Tanaka,

Thank you for submitting your manuscript to PLOS ONE. After careful consideration, we feel that it has merit but does not fully meet PLOS ONE’s publication criteria as it currently stands. Therefore, we invite you to submit a revised version of the manuscript that addresses the points raised during the review process.

Specifically, you must edit and clarify your contentions per the requests delineated by of one of the reviewers.  Please submit your revised manuscript by Aug 12 2022 11:59PM. If you will need more time than this to complete your revisions, please reply to this message or contact the journal office at plosone@plos.org. Please include the following items when submitting your revised manuscript:A rebuttal letter that responds to each point raised by the academic editor and reviewer(s). You should upload this letter as a separate file labeled 'Response to Reviewers'.A marked-up copy of your manuscript that highlights changes made to the original version. You should upload this as a separate file labeled 'Revised Manuscript with Track Changes'.An unmarked version of your revised paper without tracked changes. You should upload this as a separate file labeled 'Manuscript'.If applicable, we recommend that you deposit your laboratory protocols in protocols.io to enhance the reproducibility of your results. Protocols.io assigns your protocol its own identifier (DOI) so that it can be cited independently in the future. For instructions see: https://journals.plos.org/plosone/s/submission-guidelines#loc-laboratory-protocols. Additionally, PLOS ONE offers an option for publishing peer-reviewed Lab Protocol articles, which describe protocols hosted on protocols.io. Read more information on sharing protocols at https://plos.org/protocols?utm_medium=editorial-email&utm_source=authorletters&utm_campaign=protocols.

We look forward to receiving your revised manuscript.

Kind regards,

Alexander Rabchevsky, Ph.D.

Academic Editor

PLOS ONE

Journal Requirements:

Additional Editor Comments (if provided):

While most of the previous queries have been adequately addressed, one of the reviewers has noted that "some of the amendments in the paper are difficult to follow. i.e., pp. 24, line 399-pp 25, line 412. The ideas seem a little muddled in this section. Please work to improve the clarity of this text, and the 2 separate points you are making: 1. the limitation of the current study not using continuous naloxone together with continuous morphine, and 2. the discrepancy between the current results and previous studies with kappa opioid receptor agonists. Also consider rephrasing from "time to time." I assume you mean with acute activation of the opioid receptors versus more chronic engagement of the opioid receptors with continual morphine?"

In order to be accepted, your edited manuscript in the ensuing submission must address these issues brought up above, categorically.

Reviewers' comments:

Reviewer's Responses to Questions

**Comments to the Author**

1. If the authors have adequately addressed your comments raised in a previous round of review and you feel that this manuscript is now acceptable for publication, you may indicate that here to bypass the “Comments to the Author” section, enter your conflict of interest statement in the “Confidential to Editor” section, and submit your "Accept" recommendation.

Reviewer #1: All comments have been addressed

Reviewer #2: All comments have been addressed

2. Is the manuscript technically sound, and do the data support the conclusions?

Reviewer #1: Yes

Reviewer #2: Yes

3. Has the statistical analysis been performed appropriately and rigorously? 

Reviewer #1: Yes

Reviewer #2: Yes

4. Have the authors made all data underlying the findings in their manuscript fully available?

Reviewer #1: Yes

Reviewer #2: Yes

5. Is the manuscript presented in an intelligible fashion and written in standard English?

Reviewer #1: Yes

Reviewer #2: Yes

6. Review Comments to the Author

Reviewer #1: (No Response)

Reviewer #2: The authors addressed all of my concerns. However, some of the amendments in the paper are difficult to follow. i.e., pp. 24, line 399-pp 25, line 412. The ideas seem a little muddled in this section. Please work to improve the clarity of this text, and the 2 separate points you are making: 1. the limitation of the current study not using continuous naloxone together with continuous morphine, and 2. the discrepancy between the current results and previous studies with kappa opioid receptor agonists. Also consider rephrasing from "time to time." I assume you mean with acute activation of the opioid receptors versus more chronic engagement of the opioid receptors with continual morphine?

7. PLOS authors have the option to publish the peer review history of their article (what does this mean?). If published, this will include your full peer review and any attached files.

Reviewer #1: No

Reviewer #2: No

---

## [Author Response · Author response to Decision Letter 2]

31 Jul 2022

We would like to thank the academic editor and reviewers for carefully reading our manuscript and for the insightful comments. The comments led to an improvement of the work. Detailed responses to the reviewers are shown below. The comments from the reviewers are shown in black and our replies are shown in red. In this revised version, changes to our manuscript within the document were all highlighted by using red-colored text (with underlining) in the file named “Revised Manuscript with Track Changes”.

Reply to Reviewer #2

Reviewer #2: The authors addressed all of my concerns. However, some of the amendments in the paper are difficult to follow. i.e., pp. 24, line 399-pp 25, line 412. The ideas seem a little muddled in this section. Please work to improve the clarity of this text, and the 2 separate points you are making: 

We wish to thank the reviewer for making important suggestions. We have deleted the sentences in lines 397-412 of the revised manuscript with track changes of the previous version. We have added new sentences as described later. 

Reviewer #2:　the limitation of the current study not using continuous naloxone together with continuous morphine, 

The half-life of naloxone is approximately 10 min (Heyman JS et al., Life Sci 1986). The half-life of intrathecal morphine in humans is approximately 90 min (Sjöström S et al, Anesthesiology 1987). Therefore, investigation of the effects of continuous infusion of intrathecal naloxone during continuous infusion of morphine would be required in order to clarify the roles of opioid receptors in the morphine-induced persistent decline of locomotor function. However, it was not tested in our study. In addition, we investigated short-term effects of opioid receptor subtype-selective agonists on locomotor function (Fig. 2), but we did not investigate their long-term effects. Thus, our results did not reveal how activation of opioid receptors is involved in the persistent decline and delayed recovery of locomotor function in rats with mild SCI receiving continuous infusion of morphine for 72 hours. 

In line 397, we have added the following sentences in the Discussion section.

These results suggest that the mechanisms of persistent decline in locomotor function caused by continuous infusion of morphine are different from those of acute and transient decline of locomotor function caused by a single dose of morphine. We investigated the short-term effects of opioid receptor subtype-selective agonists on locomotor function (Fig 2), but we did not investigate their long-term effects. Continuous infusion of naloxone would be required to completely block the activation of opioid receptors during continuous infusion of morphine because the half-life of naloxone is approximately 10 min [32]. However, it was not tested in this study. Therefore, our results did not reveal in detail how activation of opioid receptors is involved in the persistent decline and delayed recovery of locomotor function in rats with mild SCI receiving continuous infusion of morphine for 72 hours.

Reviewer #2:　the discrepancy between the current results and previous studies with kappa opioid receptor agonists. 

In our study, a single dose of 0.08 �mol (30 �g) of morphine hydrochloride caused a transient deterioration of locomotor function after mild SCI via activation of �- and δ-opioid receptors but did not affect functional recovery at 14 days after mild SCI (Fig 1). In addition, we observed that continuous infusion of morphine (0.67 �mol of morphine hydrochloride administered over a period of 3 days) caused acute deterioration and persistent decline of locomotor function after SCI (Fig 3). 

It has been shown that a �-opioid receptor antagonist (WIN44,441-3) promotes functional recovery from traumatic spinal cord injury in cats (Faden AI et al., Peptides, 1985). Aceves et al. (J Neurotrauma 2017) reported that pretreatment with nor-Binaltorphimine (norBNI), a selective �-opioid receptor antagonist, blocked the adverse effects of 0.32 �mol of morphine sulfate on long-term recovery of locomotor function at 2 to 3 weeks after SCI. In addition, it has been shown that administration of GR89696, a selective �-opioid receptor agonist, does not cause acute deterioration of locomotor function but undermines recovery after SCI (Aceves M et al., Spinal Cord 2016). These results indicate that the �-opioid receptor plays a critical role in the morphine-induced attenuation of locomotor recovery. It has been suggested that a �-opioid receptor antagonist may prevent the morphine-induced persistent decline of locomotor function by reducing the extent of cell death at the site of injury [Aceves 2017].

It should be noted that morphine did not cause acute deterioration of locomotor function but delayed functional recovery in the SCI model of Aceves et al. Locomotor function at 1 day after moderate SCI in their study was lower than that in our study using rats with mild SCI. Therefore, it is likely that there was no room for further acute decline of locomotor function due to morphine administration in their model. 

 In summary, rats with mild SCI in which locomotor function was preserved to some extent were used in our study unlike in previous studies. Our results suggest that activation of �- and δ-opioid receptors is involved in the morphine-induced acute deterioration of locomotor function in animals with mild SCI, but the role of opioid receptors in morphine-induced delayed functional recovery was not examined. In contrast, previous studies have demonstrated significant roles of the �-opioid receptor in long-term recovery of locomotor function in animals with moderate SCI receiving morphine. The novelty of our study is that we showed that morphine can cause acute deterioration of residual locomotor function in animals with mild SCI via activation of opioid receptors. The results of our study taken together with the results of previous studies [6, Aceves 2016, 2017] suggest that different subtypes of opioid receptors are involved in morphine-induced acute deterioration of locomotor function and delayed recovery.

In order to make it easier to understand the differences between the previous studies and the present study, we have added the following sentences in the Discussion section in line 412.

Antagonism of the �-opioid receptor has been reported to attenuate the morphine-induced persistent decline of locomotor function by reducing the extent of cell death at the site of injury [33]. It has also been shown that an agonist of the �-opioid receptor undermines the recovery of locomotor function after a moderate degree of SCI [34]. These results indicate that the �-opioid receptor plays a critical role in the morphine-induced attenuation of locomotor recovery. It should be noted that there was a difference in the degrees of SCI in those previous studies [33, 34] and our study. Locomotor function at 1 day after SCI in the previous studies was lower than that in our study using rats with mild SCI. Morphine-induced acute deterioration of locomotor function after SCI, which was observed in our study, may occur only in the case of mild SCI in which locomotor function is preserved to some extent. The results of our study taken together with the results of those previous studies [33, 34] suggest that morphine has various impacts on residual locomotor function after mild SCI, including acute deterioration and attenuation of recovery, via different subtypes of opioid receptors.

Reviewer #2: Also consider rephrasing from "time to time."

Thank you for pointing out the confusing expression. 

Naloxone could reverse morphine-induced acute deterioration of locomotor function at 30 min after morphine administration (Figs. 2a and 4b). In contrast, a single dose of naloxone could not reverse the morphine-induced persistent decline of locomotor function at 72 hours after the start of continuous infusion (Fig. 4a). The role of opioid receptors may vary with the time elapsed after morphine administration. Therefore, we used the expression “time to time”.

Nonetheless, as the reviewer pointed out, "time to time" is vague and unclear, so we have deleted it.

Reviewer #2: I assume you mean with acute activation of the opioid receptors versus more chronic engagement of the opioid receptors with continual morphine?

The half-life of naloxone is approximately 10 min (Heyman JS et al., Life Sci 1986). The half-life of intrathecal morphine in humans is approximately 90 min (Sjöström S et al., Anesthesiology 1987). The duration of analgesia with morphine is short. Therefore, repeated or continuous administration is necessary for long-term analgesic effects. It is unlikely that a single administration of morphine causes chronic activation of opioid receptors.

In our study, a single dose of morphine was given to produce transient activation of opioid receptors, and continuous infusion of morphine for 72 hours was given to produce prolonged activation of opioid receptors.

There was a typo. We corrected it as follows.

In line 453, “U50,488H418" to “U50,488H''

We have added the following reference and adjusted the serial number of other citations.

34. Aceves M, Mathai BB, Hook MA. Evaluation of the effects of specific opioid receptor agonists in a rodent model of spinal cord injury. Spinal Cord. 2016 ;54(10): 767-777. https://doi.org/10.1038/sc.2016.28 PMID: 26927293

Again, thank you for giving us the opportunity to strengthen our manuscript with your valuable comments and queries.

---

## [Editor Report · Decision Letter 3]

3 Aug 2022

Intrathecal morphine exacerbates paresis with increasing muscle tone of hindlimbs in rats with mild thoracic spinal cord injury but without damage of lumbar a-motoneurons

PONE-D-21-29555R3

Dear Dr. Tanaka,

We’re pleased to inform you that your manuscript has been judged scientifically suitable for publication and will be formally accepted for publication once it meets all outstanding technical requirements.

Kind regards,

Alexander Rabchevsky, Ph.D.

Academic Editor

PLOS ONE
---

## [Editor Report · Acceptance letter]

5 Aug 2022

PONE-D-21-29555R3 

Intrathecal morphine exacerbates paresis with increasing muscle tone of hindlimbs in rats with mild thoracic spinal cord injury but without damage of lumbar α-motoneurons 

Dear Dr. Tanaka:

I'm pleased to inform you that your manuscript has been deemed suitable for publication in PLOS ONE. Congratulations! Your manuscript is now with our production department. 

Kind regards, 

on behalf of

Dr. Alexander Rabchevsky 

Academic Editor

PLOS ONE